

# Alteration of primary cilia and intraflagellar transport 20 (IFT20) expression in oral squamous cell carcinoma (OSCC) cell lines

Gulam Sakinah-Syed[1,2], Jia Shi Liew[1], Nazia Abdul Majid[1] and
Siti Amalina Inche Zainal Abidin[2,3]

[1] Institute of Biological Sciences, Faculty of Science, Universiti Malaya, Kuala Lumpur, WP Kuala Lumpur, Malaysia

[2] Department of Oral & Craniofacial Sciences, Faculty of Dentistry, Universiti Malaya, Kuala Lumpur, WP Kuala Lumpur, Malaysia

[3] Oral Cancer Research & Coordinating Center, Faculty of Dentistry, Universiti Malaya, Kuala Lumpur, WP Kuala Lumpur, Malaysia

Corresponding author
Siti Amalina Inche Zainal Abidin,
sitiamalina@um.edu.my

## ABSTRACT

**Background:** Aberrations in primary cilia expression and intraflagellar transport (IFT) protein function have been implicated in tumourigenesis. This study explores the relationship between the alteration of primary cilia and tumourigenesis by investigating primary cilia expression and the role of IFT20 in regulating matrix metalloproteinase 9 (MMP-9) expression in oral squamous cell carcinoma (OSCC) cell lines.

**Methods:** The frequency and length of primary cilia were determined in OKF6-TERT2 cells, HSC-2 cells, and HSC-3 cells using immunofluorescence. Additionally, primary cilia presence in non-proliferating OSCC cells was examined. OSCC cells were treated with either small interfering RNA (siRNA) negative control or siRNA targeting IFT20 for functional analysis. mRNA expression levels of IFT20 and MMP-9 were quantified using quantitative reverse transcription polymerase chain reaction (qRT-PCR).

**Results:** Results showed that HSC-2 cells exhibit abundant primary cilia when cultured in low serum media (2% serum) for 48 h, followed by serum starvation for over 72 h. No significant changes in cilia expression were observed in HSC-3 cells compared to OKF6-TERT2 cells. Ciliated cells were found in non-proliferating HSC-2 and HSC-3 cells. OSCC cells showed longer cilia than OKF6-TERT2 cells, indicating ciliary abnormalities. Changes in ciliation and cilium length of OSCC cells were accompanied by increased expression of IFT20, an intraflagellar transport protein crucial for the primary cilia assembly. However, IFT20 knockdown did not affect MMP-9 at the mRNA level in these cells.

**Conclusions.:** This study reveals the differences in primary cilia expression among OSCC cells. Furthermore, the increased abundance and elongation of primary cilia in OSCC cells are accompanied by elevated expression of IFT20. Nonetheless, IFT20 did not affect MMP-9 mRNA expression in OSCC cells.

## INTRODUCTION

Oral squamous cell carcinoma (OSCC) is the most common type of head and neck cancer, which accounts for 95% of oral cancer cases (*Moratin et al., 2021*). Globally, 900,000 cases are reported annually, with high incidence in developing countries (*Bray et al., 2018*). Despite advancements, the 5-year survival rate for OSCC patients is still low, approximately 50%, due to local invasiveness and early metastasis (*Chinn et al., 2015*). Understanding the underlying mechanism(s) is crucial for identifying novel treatment modalities and new therapeutic targets.

Primary cilium is a hair-like microtubule-based structure that senses, internalises, and converts extracellular stimuli for intracellular responses (*Gerdes, Davis & Katsanis, 2009*). Primary cilia serve as signalling hubs for certain pathways such as Hippo, Wingless/integrated (Wnt), mammalian target of rapamycin (mTOR), Sonic hedgehog (Shh), notch, transforming growth factor β (TGF-β) and platelet-derived growth factor (PDGF), which are crucial for normal cellular functions. Therefore, primary cilia dysfunction leads to various diseases such as Bardet–Biedl syndrome (*Gupta et al., 2022*), chronic kidney disease (*Park, 2018*), age-related brain diseases (*Ma et al., 2022*), heart disease (*Toomer et al., 2019*), and retinopathy (*Zhou & Zhou, 2020*). The relationship between primary cilia and cancer is complex and often depends on cell type. While it is generally observed that loss of primary cilia is associated with cancer, in certain cases, increased primary cilia can also contribute to tumourigenesis. The loss of primary cilia was demonstrated in renal (*Basten et al., 2013*), lung (*Hu et al., 2014*), prostate (*Hassounah et al., 2013*), breast (*Menzl et al., 2014*), ovarian (*Egeberg et al., 2012*), and pancreatic cancer (*Schimmack et al., 2016*). Loss of primary cilia has been reported in certain subtypes of thyroid cancer. Notably, the depletion of IFT88 and kinesin family member 3A (KIF3A) enhanced voltage dependent anion channel 1 (VDAC1) oligomerization, thereby promoting mitochondrial-dependent apoptotic cell death in differentiated thyroid cancers (*Lee et al., 2021*). These findings indicate that primary cilia are tumour suppressors in cancer development. However, the abundant presence of primary cilia in various cancers, including adenocarcinomas of the lung, colon, and pancreas, as well as in follicular lymphoma and plexiform and follicular ameloblastomas, suggests that primary cilia can also play a tumour-promoting role in specific cancer types (*Putnová et al., 2024*). A recent study demonstrated that wild-type clear renal carcinoma cell lines exhibit abundant primary cilia, which are positively correlated with von Hippel-Lindau (VHL) expression and poor prognosis (*Tian et al., 2024*). Moreover, the depletion of ciliogenesis-related genes, KIF3A and IFT88, significantly suppressed tumour proliferation and metastasis *in vitro* and *in vivo*. Similarly, primary cilia are frequently observed in pituitary neuroendocrine tumours, where their presence is associated with enhanced tumour invasion and recurrence (*Martínez-Hernández et al., 2024*). Collectively, proteins associated with cilia defects have been identified as oncogenes in multiple cancer types (*Yin et al., 2022b*).

Cilia assembly is a multistep process involving basal body maturation, ciliary membrane biogenesis, and ciliary axoneme elongation, which is controlled by the intraflagellar transport (IFT) protein. This protein complex uses kinesin (anterograde) and dynein (retrograde) as molecular motors to ensure bidirectional activity along the axoneme (*Lu et al., 2015*). The IFT proteins consist of at least 22 subunits (*Lechtreck, 2015*). IFT20, the smallest IFT subunit, is pivotal for cilia biogenesis and function (*Finetti, Onnis & Baldari, 2022*). *Follit et al. (2006)* demonstrated that the knockdown of IFT20 reduced the percentage of ciliated retinal pigmented epithelium (RPE) cells. Conversely, transfection of the IFT20-GFP expression construct into the RPE cells restored the ciliary assembly, suggesting its role in the ciliary assembly. IFT20 also possesses extraciliary functions by localising to the Golgi bodies, centrosome, early and recycling endosome for various cellular activities (*Follit et al., 2006*; *Jonassen et al., 2008*; *Keady, Le & Pazour, 2011*). Moreover, IFT20 controls lysosome biogenesis, promotes cell invasion, regulates glucose homeostasis, collagen secretion and intracellular trafficking of receptors and other signalling molecules (*Finetti et al., 2020*; *Finetti, Onnis & Baldari, 2022*). IFT20 is involved in the transportation of cell migratory enzymes from the trans-Golgi network to the plasma membrane, causing extracellular matrix (ECM) degradation and formation of invadopodia for cell invasion and migration (*Aoki et al., 2019*; *Yang et al., 2021*). Collectively, IFT20 is not only crucial in cilia maintenance, but also in non-ciliary functions.

One of the non-ciliary functions of IFT20 is the transportation of MMPs (*Nishita et al., 2017*). MMPs are a group of proteolytic enzymes that are responsible for protein degradation in the ECM (*Yin et al., 2021*). MMP-9 is the most studied MMP, and it plays a pivotal role in cancer cell invasion and metastasis. Overexpression of MMP-9 was observed in various cancers, including breast (*Yousef et al., 2014*), gastric (*Prathipaa et al., 2021*), non-small-cell lung (*Yamaguchi et al., 2004*), prostate (*Ma et al., 2016*), and OSCC (*Umashankar et al., 2021*). The activation of MMP-9 in OSCC is regulated by hedgehog (Hh) signalling pathways through the overexpression of Shh and glioma-associated oncogene homolog (Gli1) 1 (*Fan et al., 2014*). *Ocbina & Anderson (2008)* demonstrated that exposure to the Shh ligand or the active SmoA1 allele activates an Hh reporter in wild-type mouse embryonic fibroblasts (MEFs), but not in MEFs lacking IFT172 (anterograde IFT motor) or Dync2h1 (retrograde IFT motor). Subsequent studies revealed that IFT25 and IFT27 are essential for Hh signalling in MEFs (*Keady et al., 2012*; *Eguether, Cordelieres & Pazour, 2018*), while mutant IFT81 chondrocyte cells exhibited elongated cilia and altered Hh signalling (*Duran et al., 2016*). Collectively, these findings correlate defective IFT proteins with disrupted Hh signalling. The association between IFT proteins and various MMPs has been reported in previous studies. These include the association of IFT88 with MMP-13 and MMP-14 (*Coveney et al., 2018*) and the association of IFT88 with MMP-2 and MMP-9 (*Wang et al., 2017*; *Xu et al., 2019*). The association of IFT20 with MMP-9 remains unknown. Thus, the role of IFT20 in regulating MMP-9 expression in OSCC has yet to be determined.

We hypothesised that primary cilia changes are linked to the alteration of IFT20 expression and that IFT20 regulates MMP-9 expression in OSCC cell lines. This study

aims to investigate primary cilia expression in OSCC cell lines and to further determine the role of IFT20 in regulating MMP-9 expression. This study contributes to understanding how ciliary abnormalities and cilia-related proteins are involved in OSCC tumourigenesis.

## MATERIALS AND METHODS

### Cell lines

Normal oral keratinocyte cells (OKF6-TERT2) were kindly provided by Prof Rheinwald (*Dickson et al., 2000*). The cells were cultured in keratinocyte serum-free medium supplemented with 10% fetal bovine serum (FBS, Tico Europe, Netherland), Gentamicin-amphotericin (GA-1000), recombinant human epidermal growth factor (rhEG), bovine pituitary extract (BPE), insulin, hydrocortisone (KGM singleQuots, Lonza, USA) and 100 µg/ml penicillin-streptomycin (Gibco, Waltham, MA, USA). OSCC cell lines were procured from the National Institute of Biomedical Innovation (NIBIOHN, Japan). Cells were cultured in low glucose Dulbecco's Modified Eagle Medium (DMEM; Nacalai Tesque, Kyoto, Japan) supplemented with 10% FBS and 100 µg/ml penicillin-streptomycin. The media were changed every 2 days, and the cells were passaged prior to confluence. Cells were maintained in a humidified atmosphere of 5% $CO_2$ at 37 °C. OKF6-TERT2 is a normal oral keratinocyte cell line used as a control. Given that cancer is heterogeneous, HSC-2 and HSC-3 cell lines were selected to study primary cilia due to their distinct metastatic potentials. HSC-2 represents a non-invasive form of OSCC, providing a useful model to investigate the characteristics of primary cilia in non-metastatic conditions. Conversely, HSC-3 is a highly metastatic cell line known for its aggressive tumour behaviour, making it ideal for studying how primary cilia might be involved in metastatic processes. This will provide insights into how primary cilia contribute to OSCC progression.

### Culture conditions

Cells ($2 \times 10^5$ cells/well) were seeded in 6-well plates onto 13 mm coverslip in low (2%) or high (20%) serum media and were grown for 48 h. Following incubation, cells were serum-starved for 24, 48, and 72 h or cultured in low and high serum media for 5 days. For serum starvation and overgrowth conditions, media were replenished every 2 days.

For Ki67 cell proliferation, cells ($2 \times 10^5$ cells/well) were seeded onto 13 mm coverslip in low (2%) or high (20%) serum media and were grown for 48 h. Cells were further cultured in DMEM with or without serum for 24, 48, and 72 h.

### Antibodies

The primary antibodies used in this study for immunofluorescence staining included rabbit polyclonal anti-Arl13b (1:800; Proteintech), mouse monoclonal anti-γ-tubulin (1:800; Sigma-Aldrich), mouse monoclonal anti-acetylated tubulin (1:800; Sigma-Aldrich), and rabbit recombinant anti-Ki67 (1:500; Abcam). Secondary antibodies were Alexa Fluor 594-conjugated goat anti-rabbit and Alexa Fluor 488-conjugated goat anti-mouse IgG (1:1000; all from Abcam).

## Immunofluorescence

Coverslips containing adherent cells were washed in Dulbecco's phosphate buffered saline (DPBS; Nacalai Tesque) and fixed with 4% paraformaldehyde (Nacalai Tesque) for 30 min at room temperature. After the removal of paraformaldehyde, coverslips were washed with DPBS (3 × 5 min). The cells were permeabilised with 0.5% Triton X-100 (Biobasic, Canada) in DPBS for 5 min at room temperature. After several washes with DPBS, cells were blocked for non-specific antibody binding using 5% bovine serum albumin (Merck Millipore) in DPBS for 30 min at room temperature. This was followed by incubation with the primary antibodies overnight at 4 °C. Cells were then washed with DPBS (3 × 10 min) and incubated with the secondary antibodies for 2 h in the dark. The coverslips were washed with DPBS (4 × 5 min) and mounted using Fluoro-KEEPER antifade Reagent with 4′, 6-diamidino-2-phenylindole (DAPI) (Nacalai Tesque). Cells were viewed using Nikon Eclipse 90i at 40× magnification and photographed using ASI HiSKY system software. Cells were considered to have primary cilia when both ciliary membrane (arl13b) and centrosome (γ-tubulin) were visible together. The number of ciliated cells was quantified by randomly selecting five different fields per coverslip. Images obtained were analysed using ImageJ software for primary cilia frequency and length. At least 200 nuclei were counted per cell line and the frequency of primary cilia was determined by dividing the number of cilia to the number of nuclei (200 cells) in three independent experiments. Cilia lengths were quantified using the 'segmented line' tool in ImageJ software.

For Ki67 analysis, the proliferating cells were stained with Ki67 antibody while the ciliary axonemes were stained with acetylated tubulin antibody. Images were acquired from 10 randomly selected fields per cell line at 40× magnification. The percentage of ciliated cells was calculated as the number of cilia (axoneme stained by the acetylated tubulin)/number of DAPI labelled nuclei (200 cells) for each cell line.

## siRNA transfection

HSC-2 and HSC-3 cells were seeded in 6-well plates at $2.3 \times 10^5$ cells/well and were allowed to attach overnight in a humidified atmosphere of 5% $CO_2$ at 37 °C. Cells were transfected at approximately 60–70% confluency using lipofectamine RNAiMAX (Invitrogen, Waltham, MA, USA) with siRNA targeted to IFT20 or scrambled siRNA (final concentration 120 nM) in Opti-MEM medium (Invitrogen) for 48 h. Commercially available siRNA for IFT20 (Hs_705431) and Silencer™ Select Negative Control #2 siRNA were purchased from Thermo Fisher Scientific, Waltham, MA, USA.

## Quantitative real-time PCR

Total RNA was extracted from cells using the GeneMATRIX Universal RNA Purification Kit (EURX, Gdańsk, Poland) according to the manufacturer's protocol. Genomic DNA was removed using on-column DNase digestion. RNA purity and concentration were measured using a Nanodrop spectrophotometer (Thermo Fisher Scientific). An OD 260/ 280 ratio between 1.8 to 2.0 is considered a good RNA quality. Reverse transcription of 100 ng total RNA into cDNA was done using the High-Capacity cDNA reverse transcription

kit (Applied Biosystem) according to the manufacturer's instructions. The reaction mixture was started by incubating at 25 °C for 10 min and was terminated by heating at 85 °C for 5 min. Total cDNA was amplified using the TaqMan® Gene Expression Master Mix (Applied Biosystem) according to the manufacturer's protocol. Commercially available primers for IFT20 (Hs01576074_m1), MMP-9 (Hs00234579_m1) and glyceraldehyde-3-phosphate dehydrogenase (GAPDH) (Hs99999905_m1) were purchased from Applied Biosystems, USA. All reactions were prepared in technical triplicates according to the manufacturer's protocol. The reaction plates were placed on 7900 HT fast real-time PCR, Applied Biosystems, USA. The PCR amplification parameters were set as follows: 1 cycle of 95 °C for 10 min, followed by 40 cycles, each comprising two steps of 95 °C for 15 s and 60 °C for 1 min. The amplification plot and cycle threshold (Ct) values produced were analysed by using the SDS Manager programme. The relative expression of individual genes was determined using the $\Delta\Delta Ct$ method by normalising the target gene expressions to GAPDH values (endogenous control).

## Statistical analyses

Statistical analyses were performed using Statistical Package for the Social Sciences (SPSS V 20). The normal distribution of data was confirmed by the Shapiro-Wilk test. For non-parametric data, a Mann-Whitney U test was used to compare data between two groups and Kruskal-Wallis tests were applied for the data with more than two groups. Data were presented as the mean ± standard error of mean (S.E.M) from three independent experiments. The $p$-value less than 0.05 ($p < 0.05$) was set as statistically significant.

## RESULTS

### Primary cilia are present, and their number increases in HSC-2 cells but not in HSC-3 cells cultured at different conditions

Ciliogenesis is triggered by stimuli related to cell cycle arrest, including serum starvation and growth to confluence (excessive cell-cell contact) (*Moreno-Cruz et al., 2023*). Primary cilia are found during the G0/G1 phase of the cell cycle (*Fabbri, Bost & Mazure, 2019*). Thus, the confluent OKF6-TERT2, HSC-2, and HSC-3 cells were serum-starved to induce ciliogenesis. Fetal bovine serum (FBS) is routinely used in cell culture media. Different FBS concentrations directly affect cell growth, thus regulating certain biological activities such as intracellular protein synthesis (*Van der Valk et al., 2010*). This evokes the possibility that cilia expression varies across conditions where the cells were cultivated. As shown in Fig. 1A, all cell lines cultured in low serum media (2% FBS) expressed primary cilia following serum starvation at all time points. Notably, HSC-2 cells cultured in this condition showed a remarkable increase in ciliated cells at 48 and 72 h of serum starvation, with the highest cilia expression (22.3 ± 2.6%) recorded at 72 h compared to OKF6-TERT2 cells (Fig. 1B). Similarly, primary cilia were present in HSC-2 cells cultured in high serum media (20% of FBS), prior to serum starvation for over 72 h (Fig. 2A), with the highest ciliated cells recorded at 72 h was 19.0 ± 1.4% compared to OKF6-TERT2 cells (Fig. 2B). In low and high serum culture conditions, no statistically significant changes in the

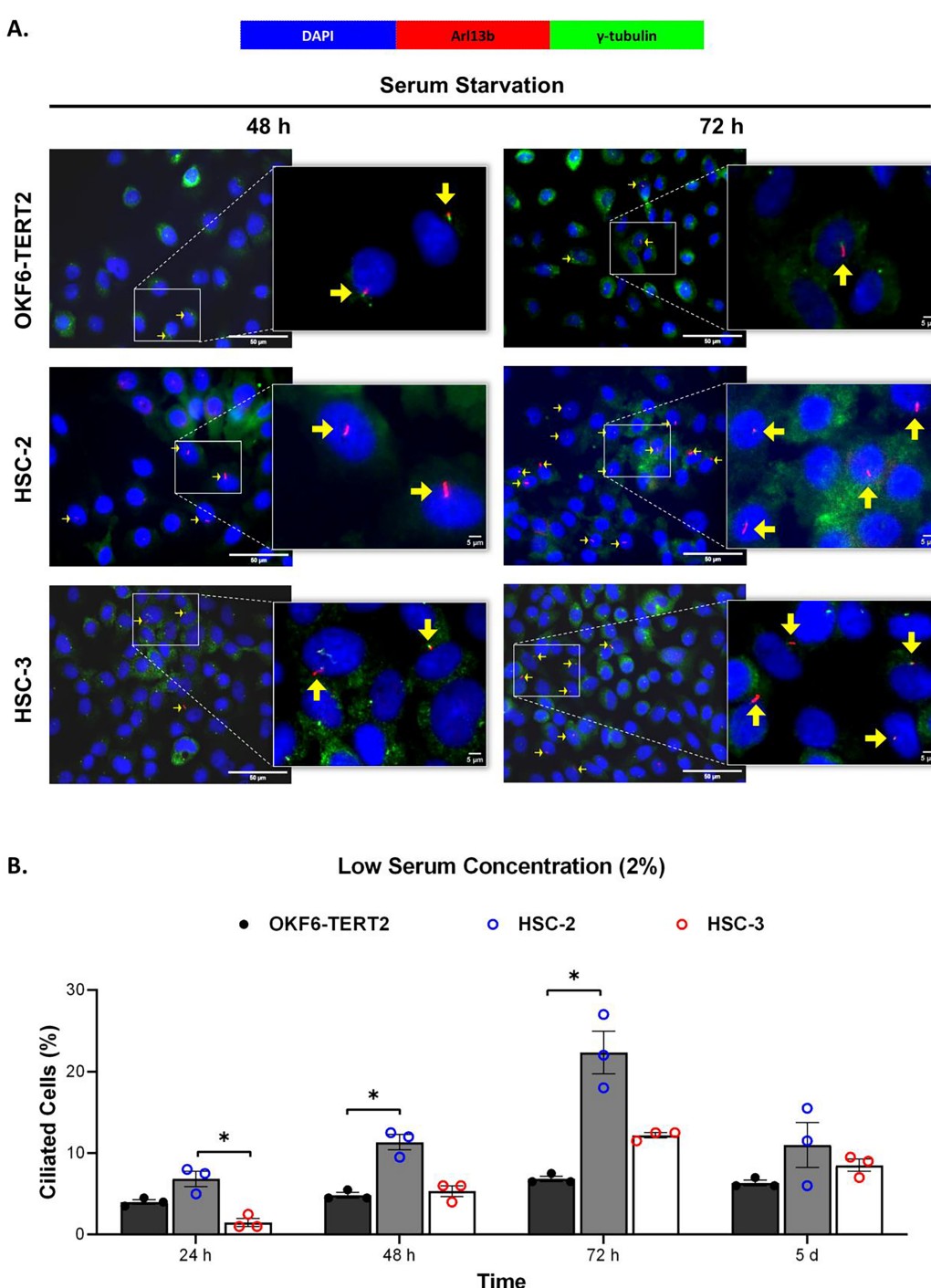

**Figure 1 The presence of primary cilia in normal oral keratinocytes and OSCC cell lines cultured in low serum media.** OKF6-TERT2, HSC-2, and HSC-3 cells were grown in media containing 2% serum for 48 h, followed by either serum starvation for 24–72 h or serum fed for 5 days. Cells were fixed and stained with antibodies against Arl13b (ciliary membrane) and γ-tubulin (basal body), as well as DAPI (nuclei). (A) The axoneme (red) and basal body (green) of primary cilia were visualised using a fluorescence microscope at 40× magnification. Scale bar: 50 μm. Arrows indicate primary cilia. (B) The percentage of ciliated cells (ciliated cells per 200 total cells) was quantified. Data represent the mean ± SEM with scattered points shown each biological repeat. Statistical analysis was performed using the Kruskal-Wallis test, followed by Dunn's multiple comparisons test. Statistical significance is indicated as *$p < 0.05$.

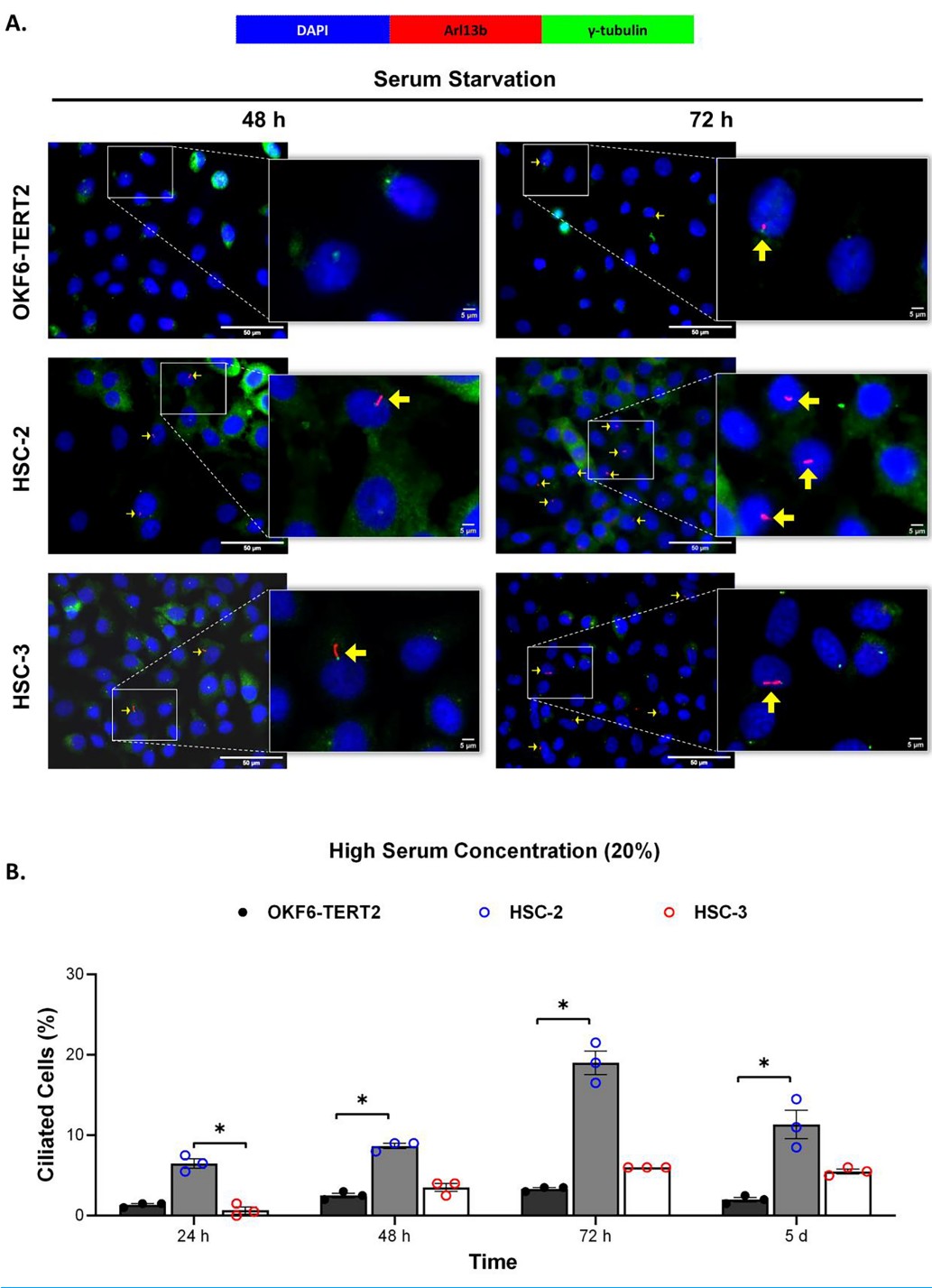

**Figure 2 The presence of primary cilia in normal oral keratinocytes and OSCC cell lines cultured in high serum media.** OKF6-TERT2, HSC-2, and HSC-3 cells were grown in media containing 20% serum for 48 h, followed by either serum starvation for 24–72 h or serum fed for 5 days. Cells were fixed and stained with antibodies against Arl13b (ciliary membrane) and γ-tubulin (basal body), as well as DAPI (nuclei). (A) The axoneme (red) and basal body (green) of primary cilia were visualised using a fluorescence microscope at 40× magnification. Scale bar: 50 μm. Arrows indicate primary cilia. (B) The percentage of ciliated cells (ciliated cells per 200 total cells) was quantified. Data represent the mean ± SEM with scattered points shown each biological repeat. Statistical analysis was performed using the Kruskal-Wallis test, followed by Dunn's multiple comparisons test. Statistical significance is indicated as *$p < 0.05$.

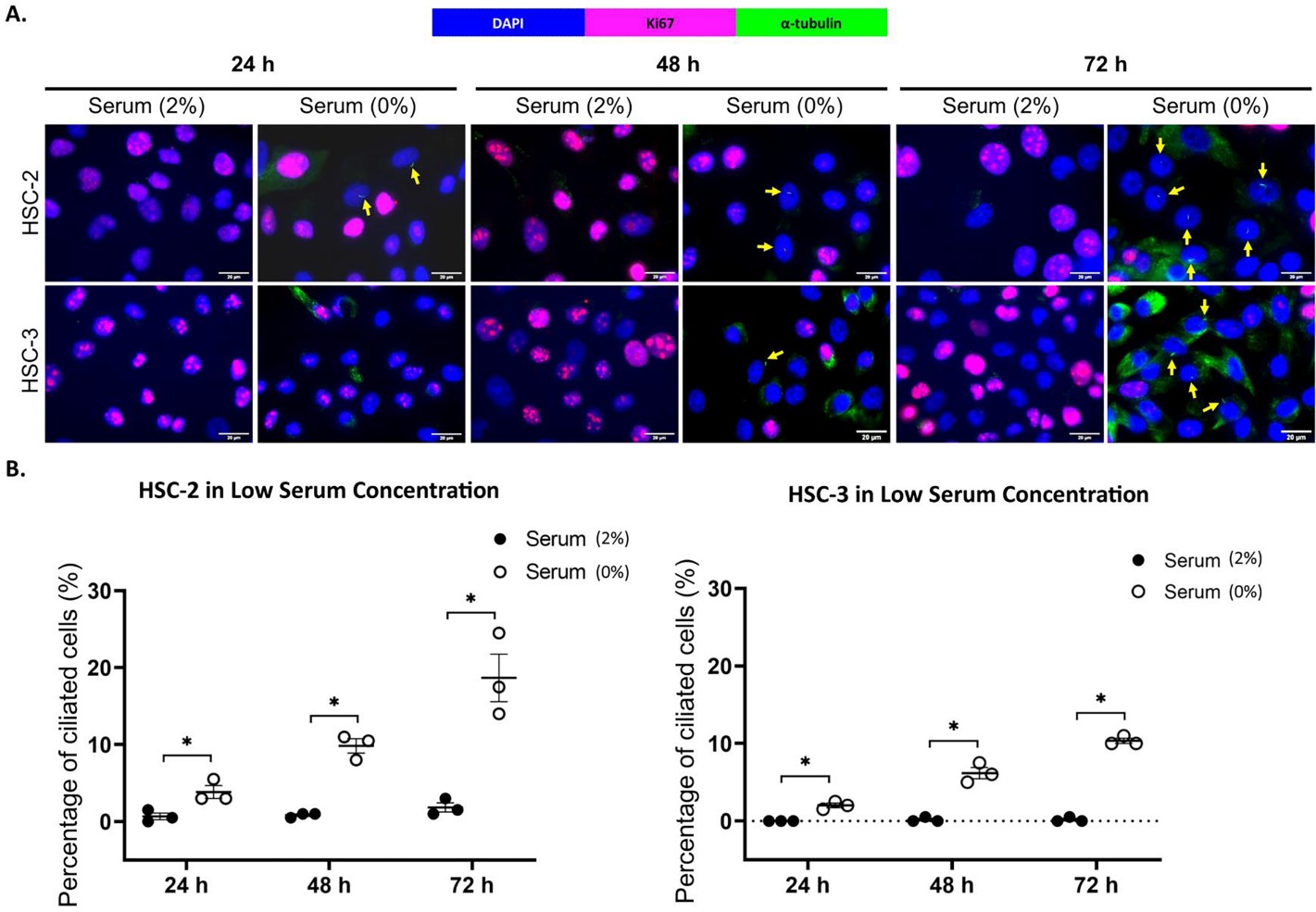

**Figure 3** **Serum starvation induces ciliogenesis in OSCC cell lines cultured in low serum media.** HSC-2 and HSC-3 cells were cultured in media containing 2% serum for 48 h. Cells were then cultured in media either with 2% serum or without serum for 24, 48, and 72 h. Fixed cells were stained with Ki67 (proliferation marker), α-tubulin (axoneme), and DAPI (nuclei). (A) The axoneme (green) of primary cilia in Ki67-negative HSC-2 and HSC-3 cells was visualised using a fluorescence microscope at 40× magnification. Proliferating cells appear magenta. Scale bar: 20 μm. Arrows indicate primary cilia. (B) Dot plots display the percentage of ciliated cells (out of 200 total cells) in HSC-2 and HSC-3. Data represent the mean ± SEM with scattered points shown each biological repeat. Statistical analysis was performed using the Mann-Whitney test. Statistical significance is indicated as *$p < 0.05$.

percentage of ciliated cells were observed in HSC-3 cells at any time points of serum starvation compared to OKF6-TERT2 cells (Fig. 2B).

### Ciliated HSC-2 and HCS-3 cells are Ki67-negative

Ki67 is commonly used as a proliferation marker in cancer. The Ki67 protein is detected in dividing cells, not in quiescent cells (G0 phase) (*Sun & Kaufman, 2018*). Primary cilia are found on the quiescent cells and are resorbed as cells re-enter the cell cycle (*Molla-Herman et al., 2008*). As shown in Figs. 3A and 4A, ciliated HSC-2 and HSC-3 cells were found in Ki67-negative cells under all culture conditions, indicating that primary cilia are reabsorbed into the cytoplasm of proliferating cells. To investigate if the increase of primary cilia in HSC-2 and HSC-3 cells is due to serum starvation, we cultured HSC-2 and

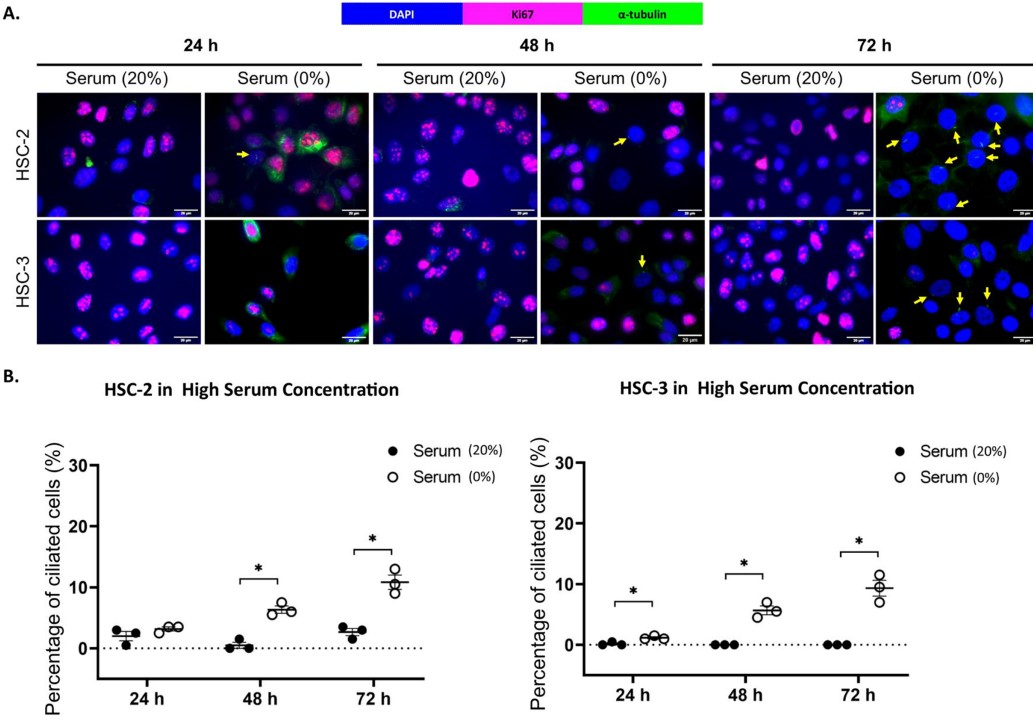

**Figure 4 Serum starvation induces ciliogenesis in OSCC cell lines cultured in high serum media.** HSC-2 and HSC-3 cells were cultured in media containing 20% serum for 48 h. Cells were then cultured in media either with 20% serum or without serum for 24, 48, and 72 h. Fixed cells were stained with Ki67 (proliferation marker), α-tubulin (axoneme), and DAPI (nuclei). (A) The axoneme (green) of primary cilia in Ki67-negative HSC-2 and HSC-3 cells was visualised using a fluorescence microscope at 40× magnification. Proliferating cells appear magenta. Scale bar: 20 μm. Arrows indicate primary cilia. (B) Dot plots display the percentage of ciliated cells (out of 200 total cells) in HSC-2 and HSC-3. Data represent the mean ± SEM with scattered points shown each biological repeat. Statistical analysis was performed using the Mann-Whitney test. Statistical significance is indicated as *$p < 0.05$.

HSC-3 cells in media containing 2% serum or without serum before serum starvation for over 72 h. We found less ciliated HSC-2 and HSC-3 cells cultured in the media containing 2% serum compared to those cultured in serum-free media across all time points (Figs. 3B and 4B).

## OSCC cell lines exhibit elongated cilium length

To determine the cilia structural changes, cilia lengths were measured in OKF6-TERT2, HSC-2, and HSC-3 cells. As shown in Fig. 5A, HSC-2 and HSC-3 cells cultured in low serum media exhibited elongated cilia compared to OKF6-TERT2 cells. HSC-2 cells cultured in low serum media showed significantly longer cilia at 48 h (3.18 ± 0.11 μm) than OKF6-TERT2 cells (Fig. 5B). Similarly, the length of cilia in HSC-3 cells increased notably, with 5 days of overgrowth condition having more elongated cilia (2.89 ± 0.19 μm) compared to OKF6-TERT2 cells.

Both HSC-2 and HSC-3 cells exhibited elongated cilia when cultured in high serum media (Fig. 6A). In HSC-2 cells, cilium length increased from 48 h to 5 days overgrowth, with the longest cilium observed at 48 h (3.24 ± 0.13 μm) (Fig. 6B). While significant

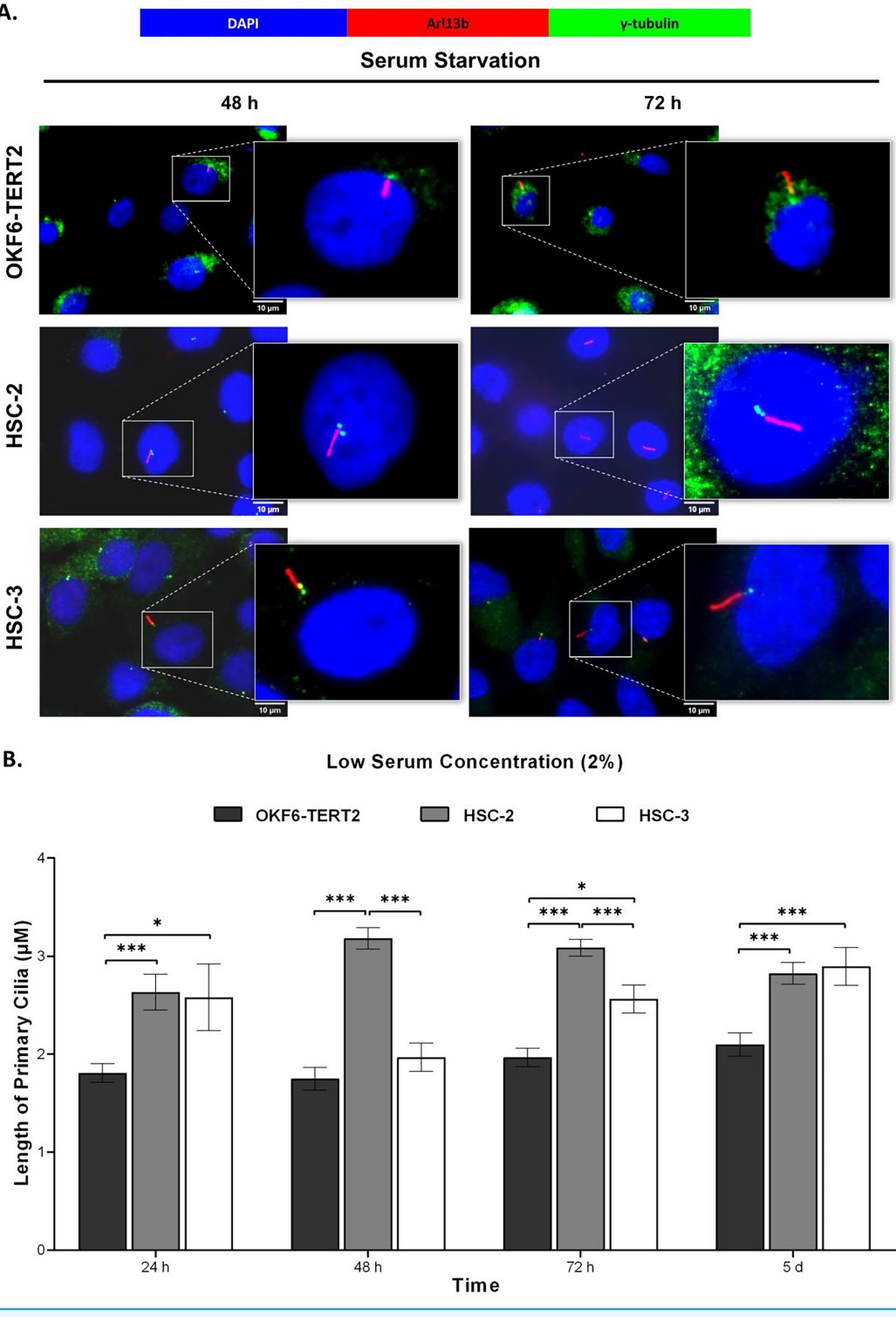

**Figure 5  Primary cilia length in normal oral keratinocytes and OSCC cell lines cultured in low serum media.** OKF6-TERT2, HSC-2, and HSC-3 cells were cultured in media containing 2% serum for 48 h, followed by either serum starvation for 24–72 h or serum fed for 5 days. Cells were fixed and stained with antibodies against Arl13b (ciliary membrane), γ-tubulin (basal body), and DAPI (nuclei). (A) The axoneme (red) of primary cilia was visualised using a fluorescence microscope at 100× magnification. A digitally zoomed image of a single cell shows the cilium length. Scale bar: 10 μm. (B) The length of primary cilia was quantified using ImageJ software. Data represent means, with error bars indicating the standard

**Figure 5** (continued)
error of the mean (*n* = 3). Statistical analysis was performed using the Kruskal-Wallis test, followed by Dunn's multiple comparisons test. Statistical significance is indicated as \**p* < 0.05 and \*\*\**p* < 0.001.

elongation of cilium was observed in HSC-3 cells at 72 h (2.55 ± 0.20 μm) and overgrowth for 5 days (2.74 ± 0.21 μm) (Fig. 6B).

## Changes in primary cilia occurrence and length were accompanied by increased expression of IFT20 in OSCC cell lines

IFT proteins are crucial for primary cilia assembly and functional competence. Next, we investigated if primary cilia changes in OSCC cell lines are associated with IFT20 expression, a subunit of the IFT complex (*Follit et al., 2006*). We evaluated the mRNA levels of IFT20 in OKF6-TERT2, HSC-2, and HSC-3 cells cultured in low or high serum media for 48 h, further serum starved for 48 and 72 h. As shown in Fig. 7A, the transcript levels of IFT20 were upregulated in HSC-2 cells cultured in low serum media at 48 and 72 h of serum starvation, with the highest IFT20 expression (2.0 ± 0.21 fold) at 72 h compared to OKF6-TERT2 cells. Similarly, the transcript levels of IFT20 were significantly upregulated in HSC-3 cells at 72 h of serum starvation (1.8 ± 0.18 fold) (Fig. 7A). No significant differences in IFT20 expression were observed in HSC-2 and HSC-3 cells cultured in high serum media at all time points (Fig. 7B).

## Knockdown of IFT20 had no effect on MMP-9 at the mRNA level

Primary cilium and its associated ciliary signalling have been linked to protease expression in fibrosis-related diseases (*Collins & Wann, 2020*). In cancer, MMP plays a key role in extracellular matrix remodelling. Accumulating evidence showed that MMP-9, a member of the MMP family, is remarkably expressed in OSCC and is correlated with OSCC progression (*Smriti et al., 2020*; *Cai et al., 2022*). These observations led to the investigation of whether IFT20 was correlated with MMP-9 expression in OSCC cells. To ascertain the role of IFT20 in regulating MMP-9 expression, HSC-2 and HSC-3 cells were transfected with siRNA negative control or siRNA IFT20 for 48 h. As shown in Fig. 8A, the mRNA levels of IFT20 were reduced by ~60% and ~90% in HSC-2 and HSC-3 cells, respectively. These findings suggest successful transient knockdown of IFT20 using siRNA in both cell lines. We then determined MMP-9 expression in transfected HSC-2 and HSC-3 cells by qRT-PCR. As indicated in Fig. 8B, no statistically significant difference in MMP-9 mRNA level was observed between the siRNA negative control and siRNA IFT20. These results show that IFT20 does not regulate MMP-9 expression in HSC-2 and HSC-3 cells.

## DISCUSSION

Accumulating evidence shows the importance of primary cilia as a signalling hub in maintaining various biological processes in normal and pathological conditions, including tumourigenesis. Primary cilia expressions vary between different cancer types. Loss of

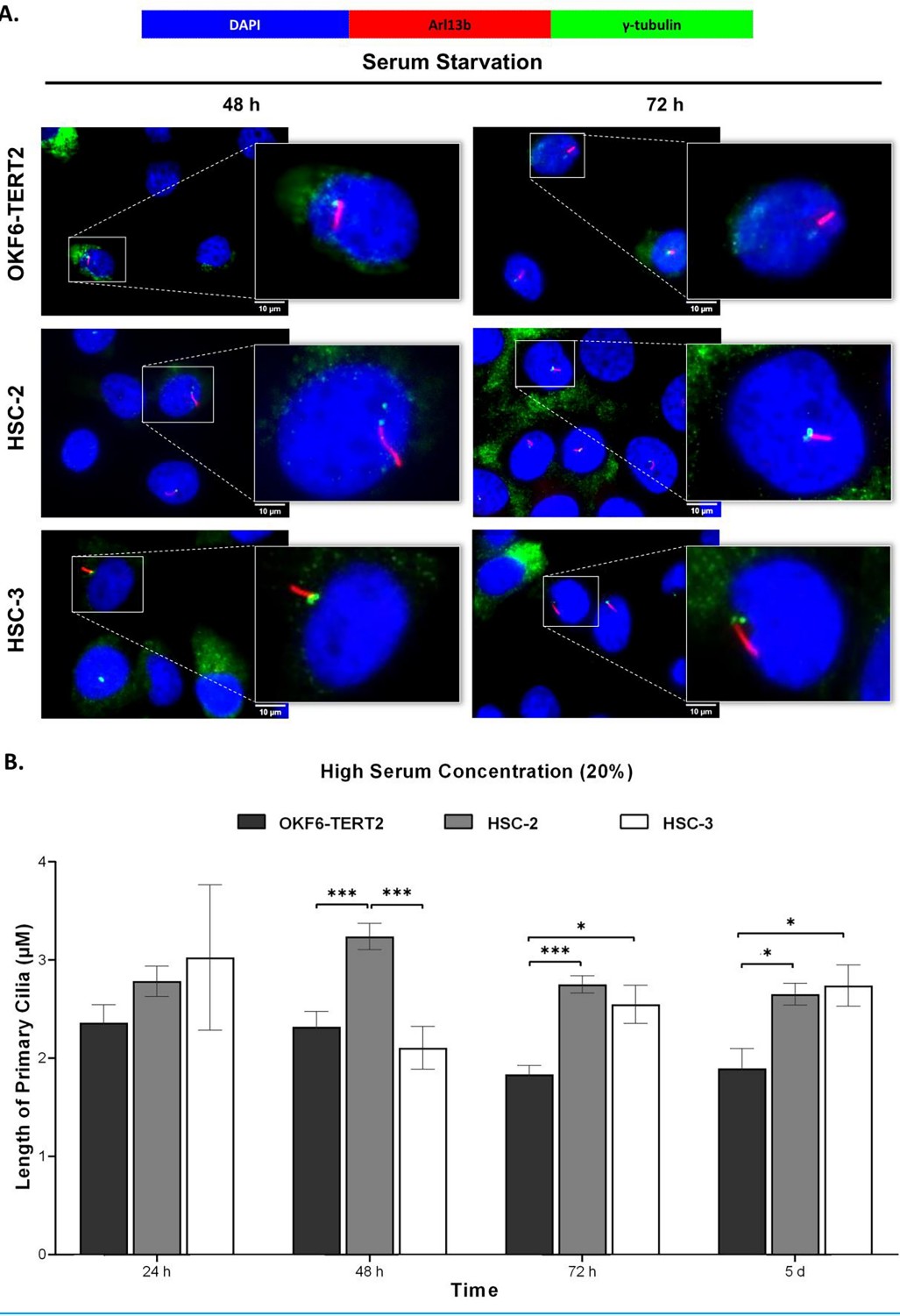

**Figure 6 Primary cilia length in normal oral keratinocytes and OSCC cell lines cultured in high serum media.** OKF6-TERT2, HSC-2, and HSC-3 cells were cultured in media containing 20% serum for 48 h, followed by either serum starvation for 24–72 h or serum fed for 5 days. Cells were fixed and stained with antibodies against Arl13b (ciliary membrane), γ-tubulin (basal body), and DAPI (nuclei). (A) The axoneme (red) of primary cilia was visualised using a fluorescence microscope at 100× magnification. A digitally zoomed image of a single cell shows the cilium length. Scale bar: 10 μm. (B) The

**Figure 6 (continued)**
length of primary cilia was quantified using ImageJ software. Data represent means, with error bars indicating the standard error of the mean ($n = 3$). Statistical analysis was performed using the Kruskal-Wallis test, followed by Dunn's multiple comparisons test. Statistical significance is indicated as *$p < 0.05$ and ***$p < 0.001$.

**A.**

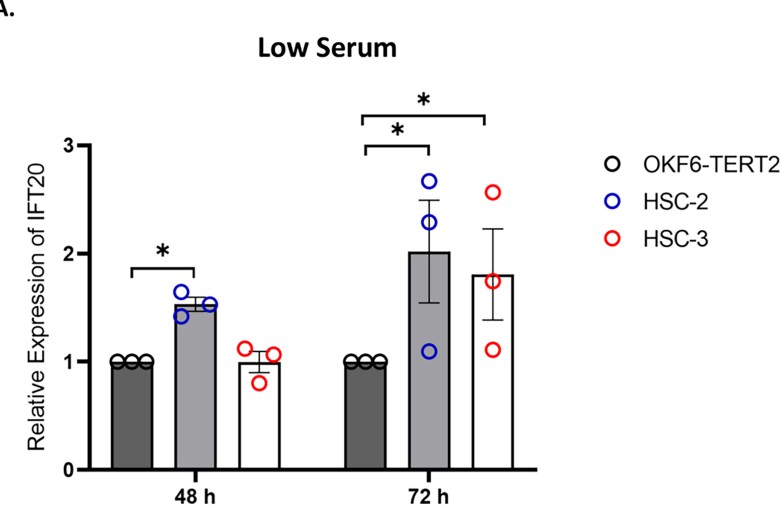

**B.**

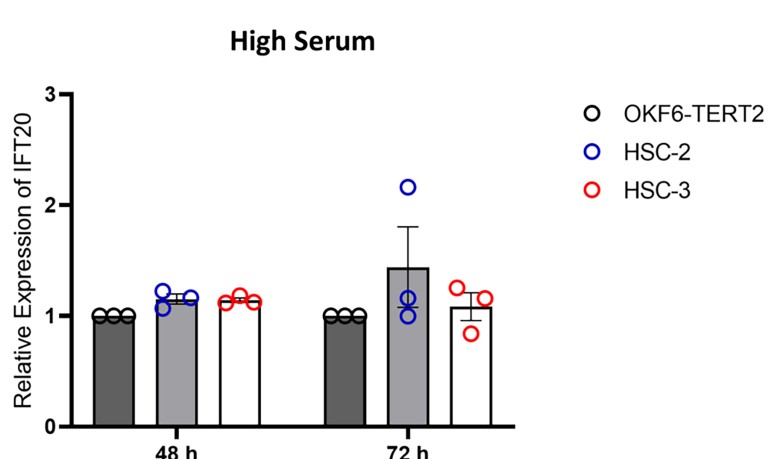

**Figure 7  IFT20 expression in OSCC cell lines cultured in low and high serum media.** OKF6-TERT2, HSC-2, and HSC-3 cells were grown in the media containing 2% or 20% serum for 48 h, followed by serum starvation for 24–72 h. RNA was extracted, reverse transcribed and subjected to qPCR to determine the mRNA level of IFT20. The bar graphs represent mRNA levels of IFT20 in HSC-2 and HSC-3 cells cultured in low serum media (A) and high serum media (B). GAPDH serves as endogenous control. Data represent the mean ± SEM with scattered points shown each biological repeat. Data were analysed using the Kruskal-Wallis test, followed by Dunn's multiple comparison test. Statistical significance is shown by *$p < 0.05$. IFT20, intraflagellar transport 20.

primary cilia was observed in various solid tumour tissues, including ovarian cancer (*Egeberg et al., 2012*), renal cancer (*Basten et al., 2013*), breast cancer (*Menzl et al., 2014*) and oral cancer (*Yin et al., 2022a*). Primary cilia were abundant in lung adenocarcinoma,

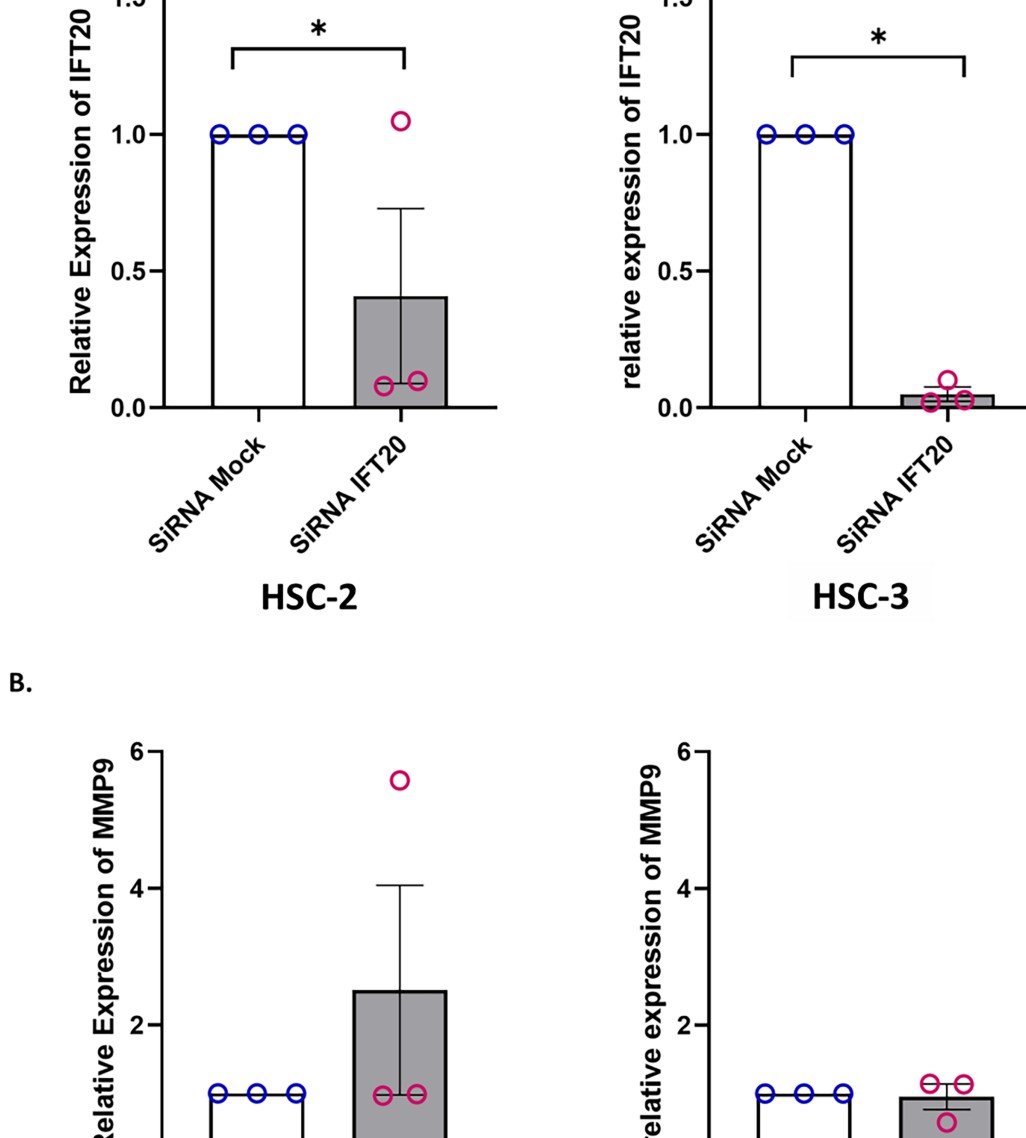

**Figure 8  IFT20 and MMP-9 expression after transient knockdown of IFT20 in OSCC cell lines.** HSC-2 and HSC-3 cells were transfected with 120 nM synthetic siRNA IFT20 or siRNA mock and incubated for 48 h. RNA was extracted, reverse transcribed and subjected to qPCR to determine the mRNA level of IFT20 and MMP-9. After transient knockdown, the downregulation of (A) IFT20 and (B) MMP-9 were determined by qRT-PCR. The mRNA levels of IFT20 and MMP-9 were normalised to GAPDH. Data represent the mean ± SEM with scattered points shown each biological repeat. Data were analysed using the Mann-Whitney test. Statistical significance is shown by *$p < 0.05$. IFT20, intraflagellar transport 20; MMP-9, matrix metalloproteinase 9; siRNA, small interfering RNA. ◺ DOI: 10.7717/peerj.18931/fig-8

colon adenocarcinoma, follicular lymphoma, and pancreatic adenocarcinoma (*Yasar et al., 2017*). Collectively, altered primary ciliogenesis contributes to the development of cancer. Most *in vitro* studies demonstrated the effects of serum starvation and growth confluence on primary cilia occurrence (*Sarkisian et al., 2014*; *Lim et al., 2015*; *Oliazadeh et al., 2017*). Only one study demonstrated the effect of serum concentrations on primary cilia prevalence in endothelial cells (*Lim et al., 2015*). FBS comprises various growth factors that possibly regulate intracellular pathways in cells. This prompts us to investigate whether different serum concentrations affect primary cilia expression in OSCC cell lines. Thus, the present study investigates primary cilia expression in HSC-2 and HSC-3 cells cultured in media containing 2% or 20% FBS for 48 h and further serum starved for over 72 h to induce ciliogenesis. The data show a marked increase in primary cilia occurrence in HSC-2 cells at various culture conditions compared to OKF6-TERT2 cells. The low levels of ciliation observed in OKF6-TERT2 cells may be attributed to their highly proliferative nature under the experimental condition. Although OKF6-TERT2 cells were cultured in keratinocyte serum-free media, they also rely on specific growth factors such as rhEGF, BPE, insulin, and hydrocortisone (media components) for attachment and growth. The presence of these growth factors, which are essential for maintaining cell viability, may suppress ciliogenesis by promoting proliferation. This could explain the reduced ciliation levels observed in OKF6-TERT2 cells under these conditions. Attempts to culture OKF6-TERT2 cells without these growth factors resulted in cell detachment from the culture plates, making it challenging to establish conditions that would promote ciliogenesis without compromising cell survival.

Cancer is often characterised by dysregulation of signalling pathways like Hh, Wnt, and mTOR, which drive uncontrolled cell proliferation and high mitotic rates. Since primary cilia are typically disassembled during the G2/M phase of the cell cycle, their loss is often linked to cancer. Primary cilia loss has been associated with increased Hh signalling in some prostate cancers (*Hassounah et al., 2013*) and Wnt signalling in melanomas (*Zingg et al., 2018*). Herein, we did not assess the expression of these signalling pathways, so it remains unclear whether disruptions in these signalling are linked to the observed increase in ciliation in HSC-2 cells. The level of ciliation may vary depending on the specific subtype of OSCC. HSC-2 cells, derived from non-metastatic OSCC, might represent a subtype where increased ciliation is a characteristic feature. Thus, further investigations are needed to determine whether this increased ciliation is associated with these signalling pathways and contributes to the unique biological properties of HSC-2 cells.

Primary cilia were present in HSC-3 cells; however, no significant differences in primary cilia frequency were observed between HSC-3 cells and OKF6-TERT2 cells across all culture conditions. The data obtained contradicts with a previous study where no primary cilia was detected in HSC-3 cells under routine cell culture or cultured in serum depletion (*Yin et al., 2022a*). Nevertheless, the duration of those cultural conditions was not sufficiently detailed. The occurrence of primary cilia was greater in HSC-2 cells than in HSC-3 cells when cultured in both low- and high-serum media, followed by 24 h of serum starvation. The difference in ciliation between these cells could be due to HSC-2 exhibits a stronger response to the stress of serum deprivation compared to HSC-3.

The primary cilium length is dynamically regulated by the assembly and disassembly of tubulin, a microtubule component (*Hilton et al., 2013*). Previous studies have shown that ciliated cells in cancer tissues or cell lines exhibit abnormal cilia lengths (*Hassounah et al., 2013*; *Yang, Roine & Mäkelä, 2013*; *Menzl et al., 2014*; *Liu et al., 2019*). We next determine the cilia morphological changes in HSC-2 and HSC-3 cells. Our findings showed that ciliated HSC-2 and HSC-3 cells exhibit longer cilia lengths than OKF6-TERT2 cells across various culture conditions. Of note, primary cilia were shorter in HSC-3 cells than in HSC-2 cells when cultured in low serum media, followed by 48 and 72 h of serum starvation. HSC-3 cells may exhibit differences in the expression or functionality of proteins essential for cilium elongation, such as IFT proteins. This is supported by our finding that HSC-3 cells show a reduced IFT20 mRNA expression compared to HSC-2 cells. Whether these data reflect the expression in the oral cancer tissue is uncertain. Thus, further work is needed to validate the *in vitro* study, which is currently underway.

Collectively, our findings demonstrate that the differential ciliogenesis observed in HSC-2 and HSC-3 cells highlight the heterogeneity of OSCC. HSC-2 cells are likely the best model for studying the role of primary cilia in OSCC biology, as they demonstrate increased ciliation, which allows for a clearer investigation of the molecular mechanisms driving cilia formation and function. However, HSC-3 cells remain valuable for understanding the consequences of less ciliated cells in metastatic OSCC. Combining insights from both cell lines provides a comprehensive understanding of the role of primary cilia in OSCC.

Primary cilia assembly is restricted to the quiescent cells (G0 phase) and is resorbed as cells re-enter the cell cycle (*Molla-Herman et al., 2008*). Ki67 is widely used in clinical settings as a proliferation marker in cancer. This is because the Ki67 protein is detected in proliferating cells and is absent in quiescent cells (*Sun & Kaufman, 2018*). Several cancer studies have reported that the primary cilia are present on the cells that are non-proliferating (*Egeberg et al., 2012*; *O'Toole et al., 2019*) and data from the present study were consistent with the previous reports, indicating that primary cilia are reabsorbed into the cytoplasm of proliferating cells. To determine whether increased primary cilia expression in OSCC cell lines is due to serum starvation, HSC-2 and HSC-3 cells were cultured in media containing 2% serum or without serum. The data showed that HSC-2 and HSC-3 cells have a higher number of ciliated cells when cultured in serum-free media compared to media with 2% serum, which aligns with findings observed by *Lim et al. (2015)*.

While most primary cilia studies in cancer focus on cancer-related signalling pathways, little is known regarding the IFT machinery. Thus, we explored the association between primary cilia and IFT20 expression in HSC-2 and HSC-3 cells. IFT20 is a part of the intraflagellar transport machinery that is essential for the primary cilia assembly. *Follit et al. (2006)* demonstrated that the knockdown of IFT20 decreases the primary cilia frequency in RPE cells. Another study revealed that the deletion of the IFT20 caused the loss of primary cilia in the progenitor cells at the hippocampal dentate gyrus region (*Amador-Arjona et al., 2011*). A study also showed that the primary cilia frequency

decreases as the degree of transformation increases during breast cancer progression (*Yuan et al., 2010*), and the loss of IFT20 promotes the migration of non-ciliated breast cancer cells (*Yang et al., 2021*). *Egeberg et al. (2012)* demonstrated a decrease in the percentage of ciliated cells in OVCAR3 and SK-OV3 (ovarian cancer cell lines) compared to wild-type ovarian cells. This decrease was not correlated with IFT20 expression. Our findings demonstrated that the alteration of ciliary frequency and length was accompanied by increased expression of IFT20 in OSCC cells.

Primary cilium and its associated ciliary signalling have been linked to modulate the expression and activity of proteases, specifically metalloproteinases in fibrosis-related diseases (*Seeger-Nukpezah & Golemis, 2012*; *Collins & Wann, 2020*). MMPs belong to a family of zinc-dependent endopeptidases that degrade most components of the ECM, generate active peptides, and activate specific growth factors, resulting in the formation of an environment that regulates the invasion and migration of cancer cells. MMP-9 has been implicated as a positive regulator for oral cancer metastasis (*Nanda et al., 2014*). The knockdown of IFT88 in extra-villous trophoblast cells decreases the expression of MMP-2 and MMP-9 which in turn reduces the cell motility, migration, and invasion (*Wang et al., 2017*). Meanwhile, IFT20 promotes cancer cell migration and invasion *via* vesicular transport of signalling molecules and transmembrane protein (*Finetti et al., 2009*; *Finetti & Baldari, 2013*; *Finetti et al., 2014*), polarised localisation of Golgi complex (*Noda et al., 2016*), and regulation of microtubule dynamics (*Finetti et al., 2009*; *Nishita et al., 2017*). Previous studies show that IFT20 is involved in the invasion of human osteosarcoma cells by modulating the intra-Golgi transport of surface-exposed membrane type1-MMP (MT1-MMP), that is responsible for matrix degradation in osteosarcoma (*Castro-Castro et al., 2016*; *Yang et al., 2021*). However, the role of IFT20 in modulating MMP-9 expression remains unknown. The functional role of IFT20 in regulating the MMP-9 expression in OSCC cell lines is investigated in the current study. The results, however, demonstrated that the knockdown of IFT20 does not affect the expression of MMP-9 at mRNA level.

In conclusion, our findings demonstrate that increased ciliogenesis in HSC-2 cells and cilia elongation in both HSC-2 and HSC-3 cells are accompanied by elevated expression of IFT20. To the best of our knowledge, this is the first study to explore the association of IFT20 with OSCC, highlighting a potential link between primary cilia and the regulation of IFT proteins in this cancer type. These findings suggest that IFT20 may play a role in OSCC biology, potentially offering new avenues for understanding the molecular mechanisms underlying tumour initiation and progression. The observed differences in primary cilia dynamics between HSC-2 and HSC-3 cells further highlight the role of ciliogenesis in contributing to tumour heterogeneity. HSC-2 cells exhibited significantly higher ciliogenesis and longer cilia compared to HSC-3 cells, suggesting that variations in ciliary assembly and length could reflect distinct biological characteristics such as cancer-initiating mutations, altered molecular pathways and differential responses to environmental stimuli. While these findings provide novel insights, they remain preliminary and warrant further investigation to validate their relevance *in vivo* and to elucidate the mechanisms underlying these observations. The association between IFT20

expression, ciliary dynamics, and tumour heterogeneity also opens intriguing possibilities for ciliotherapy, a therapeutic strategy aimed at targeting primary cilia and their associated proteins. However, additional studies are needed to explore these possibilities and determine the translational potential of ciliotherapy in OSCC.

This study has several limitations that warrant further investigation. First, it is too premature to link between IFT20 and primary cilia changes in OSCC, despite observing increased abundance and elongation of primary cilia in OSCC cells along with elevated expression of IFT20. Further cilia specific analyses are needed to validate its role in ciliogenesis in OSCC cells. These include the localisation of IFT20 in OSCC cells and primary cilia expression in IFT20-knocked-down cells. Second, although we demonstrated successful inhibition of IFT20 at the mRNA level, further protein analysis is essential to validate this finding. Furthermore, using two or more siRNA sequences for IFT20 knockdown will enhance the reproducibility and reliability of our results. Third, while our data indicates that IFT20 is not associated with MMP-9 expression at the mRNA level, this evidence alone is insufficient to fully understand the role of IFT20 in regulating MMP-9. Therefore, further protein analyses such as immunoblotting and gelatin zymography are necessary to support this evidence.

## CONCLUSIONS

Our data demonstrate that increased primary cilia expression and structural changes in the length of primary cilia in OSCC cells are associated with aberrant expression of IFT20. These data provide insights into the involvement of IFT20 in ciliogenesis within OSCC cells. Hence, further investigation is required to elucidate the role of IFT20 in the ciliogenesis of OSCC. We have demonstrated that IFT20 does not regulate MMP-9 expression in OSCC cells. Our data do not support an association between IFT20 and matrix metalloproteinase, particularly MMP-9, suggesting that IFT20 may have a more specialised role that does not involve the regulation of MMP-9. However, more experiments are needed to confirm these findings.

## ACKNOWLEDGEMENTS

The authors are grateful to Prof. Rheinwald for providing the OKF6-TERT2 cell line.

### Funding

This work was funded by the Ministry of Higher Education (MoHE), Malaysia [Fundamental Research Grant Scheme (Grant Number: FRGS/1/2019/SKK14/UM/02/12)] and the Universiti Malaya Research Fund (Grant number: ST043-2020). The funders had no role in study design, data collection and analysis, decision to publish, or preparation of the manuscript.

## Grant Disclosures

The following grant information was disclosed by the authors:
Ministry of Higher Education (MoHE), Malaysia.
Fundamental Research Grant Scheme: FRGS/1/2019/SKK14/UM/02/12.
Universiti Malaya Research Fund: ST043-2020.

## Competing Interests

The authors declare that they have no competing interests.

## Author Contributions

- Gulam Sakinah-Syed performed the experiments, analyzed the data, prepared figures and/or tables, authored or reviewed drafts of the article, and approved the final draft.
- Jia Shi Liew performed the experiments, analyzed the data, prepared figures and/or tables, and approved the final draft.
- Nazia Abdul Majid conceived and designed the experiments, authored or reviewed drafts of the article, and approved the final draft.
- Siti Amalina Inche Zainal Abidin conceived and designed the experiments, authored or reviewed drafts of the article, and approved the final draft.

## Data Availability

The raw data obtained are available in the Supplemental File.

## Supplemental Information

Supplemental information for this article can be found online at http://dx.doi.org/10.7717/peerj.18931#supplemental-information.

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
