# Peer review of "Alteration of primary cilia and intraflagellar transport 20 (IFT20) expression in oral squamous cell carcinoma (OSCC) cell lines"

_PeerJ, doi:10.7717/peerj.18931_

## Round 0.1 · original submission · Major Revisions

Please address concerns of two reviewers and amend manuscript accordingly.

Reviewer 1 ·

Basic reporting

The manuscript titled "Alteration of primary cilia and IFT20 expression in oral squamous cell carcinoma (OSCC) cell lines" presents a study investigating the relationship between primary cilia expression and the role of IFT20 in regulating MMP-9 expression in OSCC cell lines. The English used in the manuscript is generally clear and professional, with appropriate field background and context provided. The article structure conforms to discipline norms, and the figures and tables are relevant and of high quality. Raw data is supplied, which aligns with PeerJ policy.

Suggested improvements:
The authors should ensure that all terms and abbreviations are defined at their first mention, especially those that may not be familiar to readers outside the immediate field.
While the figures are of high quality, the authors should ensure that all figure legends are detailed enough to allow for the figures to be understood without referring to the main text.
It would be beneficial to include a supplementary file or additional methods section detailing the statistical tests used for each experiment, as this information is crucial for the reproducibility and understanding of the study's findings.

Experimental design

The research question addressed in the manuscript is well-defined and relevant, as it aims to fill a knowledge gap regarding the role of IFT20 in OSCC cell lines. The investigation appears to be rigorous and performed to a high technical and ethical standard. The methods are described with sufficient detail to allow for replication.

Suggested improvements:
The authors should provide more information on the selection of specific cell lines (OKF6-TERT2, HSC-2, and HSC-3) and justify their relevance to the study's aims.
It would be helpful if the authors could elaborate on the choice of siRNA sequences used for IFT20 knockdown and whether these sequences have been validated in previous studies.
The study would benefit from additional experiments to validate the findings at the protein level, as mRNA expression alone may not fully represent the functional impact of IFT20 on MMP-9 expression.

Validity of the findings

The study provides underlying data that appear to be robust and statistically sound. The conclusions drawn by the authors are well-stated and limited to the supporting results, which is in line with the editorial criteria.

Additional comments

The manuscript would benefit from a more detailed discussion on the implications of the findings for the understanding of OSCC biology and potential therapeutic strategies. Additionally, the authors should consider expanding the introduction to provide a broader context of the role of primary cilia in cancer and how the current study advances the field. Finally, the authors should ensure that the manuscript adheres to the latest guidelines for reporting preclinical research, including the ARRIVE guidelines, to enhance the transparency and reproducibility of the study.

·

Basic reporting

The paper was written in Avery clear language with proper literature's review and a sufficient background about the experiment.
All tables & figure were depicted properly with proper labelling.
No need for any further corrections.

Experimental design

The research is within the aims & scope of the journal.
The research question is well defined and very meaningful and identified the knowledge gap.
Methods was described with sufficient detail & the experiment is replicable.

Validity of the findings

The work is novel in dealing with specific IFT20 gene and its effect on ciliogenesis.
The tables and the data presented in the table were based on very good statistics.
The conclusion is well stated and answered the primary question.

Additional comments

I would like to congratulate the authors for writing this masterpiece of research.

Reviewer 3 ·

Basic reporting

The level of english is adequate. The references suffice. The data, necessary for understanding, has been provided, as well as figures of the research involved. Nice discussion and introduction.

Experimental design

The research is original, the research questions are adequately answered. The methods are thoroughly explained.

Validity of the findings

The findings are well supported, and the replication of the process is feasible, since the methods were explained. The conclusions are well stated.

Reviewer 4 ·

Basic reporting

In this manuscript by Sakinah et al. the authors highlight that there is a current gap in the reported knowledge of the relationship between MMP-9 and IFT20. Meanwhile, the link between other metalloproteinases and IFT components have been reported previously. They characterize two OSCC cell lines (HSC-2, and HSC-3) in contrast with normal oral epithelial cell line (OKF6-TERT2), in their ability to produce primary cilia and further test if ciliary components’ presence, here IFT20, will affect the expression of MMP-9.

While the body of work is at large valid and suitable for the journal, there are multiple points that need to be addressed before considering it for acceptance.

1.1. The Peer J author guidelines instruct to report all statistical results with the exact p-value and in the figures and legends, these are shown by asterisk or "p<0.05".

1.2. Line 31-32: It is a bit confusing, since the authors have not shown how the cells are ciliated in multiple serum conditions, as per their methods section, the cells were only cultivated prior in 2% serum media.

1.3. Line 146-147: The text needs to be corrected, because ARL13B is not an axonemal protein, instead it’s associated with the ciliary membrane (PMID: 21976698).

1.4. For both Figure 1A and 2A I think it would be helpful to introduce a zoomed in panel to observe the primary cilium in greater detail, because they are very hard to discern in the current pictures. Much like they did in Figure 3 and 4.

1.5. Lines 205-206: The presented results in the current form are a bit confusing. The text and Figures 1 and 2, should mention that the low or high serum conditions are prior to the serum starvation. This is clarified only in the methods section, but it’s not easy to understand from the figures or the text. Since the cells were cultured in low (2%) or high (20%) serum media prior to serum starvation, there is no data of the ciliation percentage in 2% FBS conditions.

1.6. Figure 5A, 6A: Would benefit greatly from having the legend of the labelled proteins in the figure. We see three different colors and they are mentioned in the figure legend, but it’s not visually communicated what these counterstains are for like in Figures 1A, 2A.

1.7. For clarity reasons, I suggest the authors move the section (and figures) regarding the Ki67-negative cells data before the section characterizing the cilium length. In this way, the ciliary presence experiments will be addressed one after the other, which would improve the readability.

Experimental design

The authors define clearly define their scientific question. They further proceed to test their hypothesis with a series of experiments that are adequate, but leaving some experimental gaps. In order to improve the manuscript and bring it up to the Peer J standards, the points bellow should be addressed.

2.1. The authors mention in the methods section that they have used both ARL13B and acetylated tubulin antibodies to detect the primary cilium. Although, no data using the acetylated tubulin is presented. Is there a difference between the ciliation detected using the two antibodies? Certain cell lines have difficulties with the ARL13B enrichment of the primary ciliary membrane, and the axoneme detection by the tubulin staining is preferred for proper quantification of the ciliation.

2.2. The use of Cytochalasin D to induce ciliogensis in presence of serum is a routine work in the field of the primary cilia (PMID: 33392209). Did the authors consider using pharmacological treatment to induce or improve the ciliogenesis of the cell lines?

2.3. Figures 5B, 6B: Are a little bit hard to follow. Instead of Serum (+) and Serum (-), I would suggest to replace it with Serum (2%) vs Serum (0%) in Figure 5, and in Figure 6. Serum (20%) vs Serum (0%) for clarity. Simultaneously, other cell lines used in primary cilia studies, such as NIH 3T3s, require 0.1% of serum in order to avoid cell detachment and improve the ciliation. Would it be possible to pick only one timepoint that shows the best efficiency and test the ciliation in multiple serum concentration conditions such as: 0%, 0.1%, 2%, 5% and then normal 10%? By this data, the characterization of the ciliation in HSC-2 and HSC-3 cells would be greatly improved.

Validity of the findings

The authors of the present manuscript conclude that all three cell lines can form primary cilia under specific culture conditions prior to synchronization by serum depletion, but it is increased in HSC-2 and HSC-3.The authors correlate the increased ciliation with the increased expression of IFT20 and observe that this does not affect the MMP-9 mRNA levels.

Reviewer 5 ·

Basic reporting

The manuscript is likely to meet the journal's standards, but it would benefit from revisions.

Experimental design

The manuscript is likely to meet the journal's standards, but it would benefit from revisions.

Validity of the findings

The manuscript is likely to meet the journal's standards, but it would benefit from revisions.

Additional comments

Kindly review the attached comments carefully.

Annotated reviews are not available for download in order to protect the identity of reviewers who chose to remain anonymous.

---

## Round 0.2 · accepted · Accept

All issues pointed by the reviewers were adequately addressed and the revised manuscript is acceptable now.

Reviewer 1 ·

Basic reporting

Language and writing skills:
The manuscript is well written and uses professional, clear, unambiguous language.
The background section is comprehensive and provides a good background for the study.
Literary background:
Make appropriate reference to relevant literature. The authors cite recent research to support the rationale of their study.
However, some sentences in the introduction could benefit from more precise citation placement (for example, discussing the dual role of primary cilia in tumorigenesis).
Charts and data display:
The charts are of high quality, well labeled and well described.
Raw data has been provided to meet journal requirements.
Structure and format:
The manuscript follows PeerJ's structural guidelines.
Table legends and methods are detailed enough to be easily copied.

Experimental design

Originality and Relevance:
This is original primary research that falls well within the journal’s scope.
The study addresses a meaningful knowledge gap by exploring the role of primary cilia and IFT20 in OSCC.
Methodological Details:
The methods are described with sufficient detail to allow replication. For example, immunofluorescence and qRT-PCR protocols are well-detailed.
It is suggested that further validation of IFT20 knockdown efficiency at the protein level (e.g., via Western blotting) be performed in future studies, as acknowledged by the authors.
Experimental Rigor:
The experimental design is robust. The inclusion of multiple OSCC cell lines (HSC-2 and HSC-3) strengthens the study's conclusions.
The authors' use of both low and high serum conditions to study ciliogenesis is commendable.
Cell Line Provenance:
Cell lines are properly described, with references provided for their origins.

Validity of the findings

Data Robustness:
The findings are robust, supported by appropriate statistical analyses.
The authors have appropriately interpreted their data, but the conclusion that IFT20 is not involved in MMP-9 regulation at the mRNA level could benefit from additional clarification about the need for protein-level studies.
Relevance of Results:
The conclusions are well-linked to the data, addressing the original research question about the association between primary cilia, IFT20 expression, and OSCC tumorigenesis.
Limitations:
The authors acknowledge the study's limitations, including the need for further protein analysis and ciliary localization studies. These are reasonable and do not detract from the validity of the findings.

Additional comments

The manuscript is a valuable contribution to understanding ciliogenesis and IFT20's role in OSCC. It demonstrates strong experimental rigor and addresses a critical knowledge gap.
The authors have thoroughly discussed the implications of their findings, including their potential application in developing therapeutic strategies targeting ciliogenesis.

Reviewer 4 ·

Basic reporting

The authors have greatly improved the reviewed manuscript.

Experimental design

no comment

Validity of the findings

The adjustments done to the text and figures brought more clarity in communicating their findings.

Reviewer 5 ·

Basic reporting

no comment

Experimental design

no comment

Validity of the findings

no comment

Additional comments

The authors have adequately addressed all my concerns and implemented the required changes to the manuscript.